# Carbon fixation from mineral carbonates

Brandon S. Guida[1], Maitrayee Bose[2] & Ferran Garcia-Pichel[1,3]

Photoautotrophs assimilate oxidized carbon obtained from one of two sources: dissolved or atmospheric. Despite its size, the pool of lithospheric carbonate is not known to be a direct source for autotrophy. Yet, the mechanism that euendolithic cyanobacteria use to excavate solid carbonates suggests that minerals could directly supply $CO_2$ for autotrophy. Here, we use stable isotopes and NanoSIMS to show that the cyanobacterium *Mastigocoleus testarum* derives most of its carbon from the mineral it excavates, growing preferentially as an endolith when lacking dissolved $CO_2$. Furthermore, natural endolithic communities from intertidal marine carbonate outcrops present carbon isotopic signatures consistent with mineral-sourced autotrophy. These data demonstrate a direct geomicrobial link between mineral carbonate pools and reduced organic carbon, which, given the geographical extent of carbonate outcrops, is likely of global relevance. The ancient fossil record of euendolithic cyanobacteria suggests that biological fixation of solid carbonate could have been relevant since the mid-Proterozoic.

[1] School of Life Sciences, Arizona State University, Tempe, AZ 85287, USA. [2] School of Earth and Space Exploration, Arizona State University, Tempe, AZ 85287, USA. [3] Center for Fundamental and Applied Microbiomics, Biodesign Institute, Arizona State University, Tempe, AZ 85287, USA. Correspondence and requests for materials should be addressed to F.G.-P. (email: ferran@asu.edu)

The biological reduction of inorganic to organic carbon, or carbon (C) fixation, is arguably the most important biogeochemical transformation on Earth, bridging the geological and biological realms, and sustaining all global biomass. The three existing inorganic C pools are directly linked through abiotic processes[1], but only dissolved C from the bulk aqueous medium or gaseous pools from the atmosphere can serve as substrate for autotrophs[2, 3]. At some $10^7$–$10^9$ GT of C, lithospheric carbonates represent, by far, the largest reservoir in the global carbon cycle[4, 5] and a practically inexhaustible potential C source. Euendolithic cyanobacteria are widespread photoautotrophs that thrive in intimate contact with carbonate substrates[6], boring into the exposed mineral surface[7]. Recent studies using the model euendolith *Mastigocoleus testarum* strain BC008 have helped unravel the physiological mechanisms of boring, an otherwise geochemically paradoxical process[8]. The current physiological model proposes that carbonate dissolution occurs via $Ca^{2+}$ removal from the boring front, followed by cell-to-cell transport and eventual ion extrusion at the substrate surface[9, 10]. During this process, protons are counter-transported towards the boring front, an action that likely results in the localized formation of dissolved $CO_2$ from $CO_3^{2-}$ released during carbonate dissolution in the interstitial space between cell and mineral inside of the solid. This would theoretically allow for fixation of mineral-sourced C by the excavating organism. Thus, the hypothesis that euendolithic cyanobacteria may be fixing the carbonate released during excavation of their own habitat is attractive for the following reasons: on one hand, it completes the geomicrobial action on the substrate, and on the other, it provides endoliths with a competitive advantage over photosynthetic epiliths, which may suffer from dissolved inorganic C (DIC) limitation as their biofilms thicken[11, 12]. Because carbonates have varying $\delta^{13}C$ signatures, often distinct from that of their local bulk seawater DIC[13], one can then use stable isotope analyses to track C sources in euendoliths. We searched for evidence for the use of mineral sources of C in endolithic autotrophy, and for a role of external C limitation in this process. By showing that mineral substrate isotopic signature is mirrored in the isotopic signature of biomass we provide compelling evidence for direct fixation of mineral derived carbon into endolithic biomass, in culture and nature. We also show that external DIC limitation enhances the propensity with which our laboratory model strain bores into calcite.

## Results

**DIC limitation enhances endolithic infestation.** *M. testarum* BC008 can grow in either boring mode, producing endolithic biomass, or in a non-boring mode (producing planktonic or suspended biomass)[14]. Endolithic filaments can also grow out into the liquid medium, still attached to the substrate, as what we call benthic biomass. We reasoned that if BC008 fixed mineral carbon, cultures subjected to prolonged DIC limitation would show higher endolithic biomass yields than cultures with no imposed limitation. We tested this directly, growing cultures for 4 months in hermetically sealed vessels (DIC-limited, containing only 1.05 mg DIC), compared with controls grown open to the atmosphere. Cultures were inoculated using either planktonic biomass or with calcite chips seeded with incipient endolithic biomass, to assess the "choice" to initiate boring under limitation vs. the "choice" to leave the substrate in the absence of DIC. Expectedly, non-limited cultures grew more than DIC-limited cultures ($8.4 \pm 1.2$ vs. $5.1 \pm 0.8$ mg). But in the latter, yields exceeded the theoretical yield from the available 1.05 mg of DIC (2.6 mg of dry biomass), indicating that calcite C had to have been the additional source. Non-limited cultures yielded less endolithic and more benthic biomass (Fig. 1a, b, f). Planktonic biomass from DIC-limited cultures displayed pigment bleaching, symptomatic of physiological stress, but not that from controls (Fig. 1c, d). Yet, endolithic and benthic biomass remained unbleached in all cultures (Fig. 1b), indicating that access to solid carbonates sufficed to relieve DIC limitation symptoms. Highest

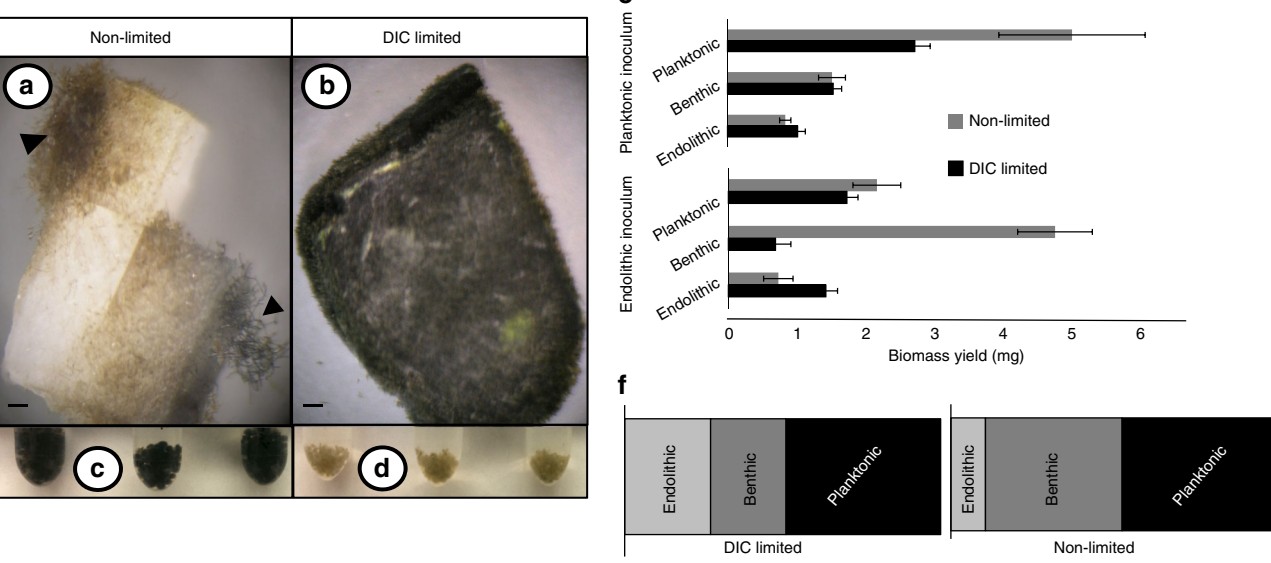

**Fig. 1** Effects of dissolved inorganic carbon limitation on *M. testarum* BC008 cultured biomass. **a, b** (scale bars = 0.5 mm) both show calcite chips with endolithic and benthic biomass growth. **c, d** show pelleted planktonic biomass (each tube is an independent replicate). Growth under dissolved inorganic carbon (DIC) limitation produced virtually complete endolithic surface infestation, short benthic outgrowth (**b**), and relatively little, bleached planktonic biomass (**d**). Growth without DIC limitation resulted in asymptomatic, abundant planktonic growth (**c**), but more restricted endolithic yield (**a**). Biomass yield obtained from each growth mode is displayed as a function of initial inoculum type (**e**). **f** shows relative distribution of the average biomass yield by growth mode under DIC-unlimited and -limited growth conditions. Error bars equal one s.d.

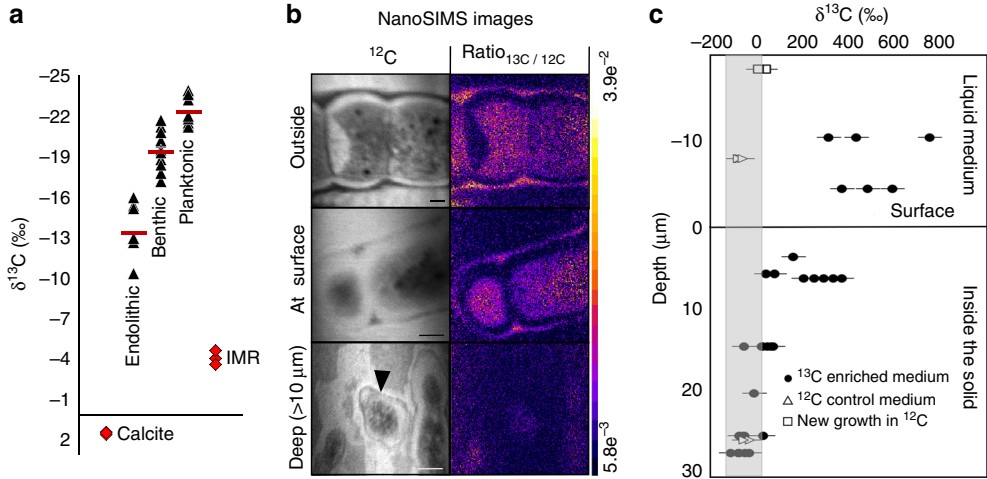

**Fig. 2** Stable isotopic evidence for carbonate C fixation. **a** Natural $\delta^{13}C$ abundance in endolithic, benthic and planktonic cultured biomass of *M. testarum* strain BC008 obtained via CF-IRMS (Red bars indicate means of at least five independent cultures). All means were significantly different (ANOVA, $p < 0.06$) from one another. The $\delta^{13}C$ of potential sources of C (medium DIC and calcite substrate) are red closed diamonds. **b, c** correspond to cultures incubated with $^{13}C$-DIC enriched medium, and started with very lightly bored inocula (Supplementary Fig. 2), so that the virtual totality of the biomass grew in the presence of the tracer. **b** Exemplary NanoSIMS images of cells analyzed for isotopic composition of organic C and obtained from different positions in a boring bed (scale = 2 μm), with full quantitative data obtained from such images as a function of depth shown in **c** (Deep$^{12}C$ image—black arrowhead indicates a single cell of several observable in this image). Also included in **c** are control cells grown in medium without tracer (triangles), and cells grown after removal of the tracer (squares); the gray box represents $\delta^{13}C$ values indistinguishable from background using this technique. Error bars equal one s.d. IMR = standard marine BC008 growth medium. DIC: dissolved inorganic carbon

yields of planktonic biomass ($p < 0.01$; Fig. 1e) were expectedly obtained from non-limited cultures inoculated with planktonic biomass, and the highest endolithic yield from limited cultures inoculated with endolithic biomass ($p < 0.05$). Limited cultures consistently yielded more endolithic biomass and bored deeper (Supplementary Fig. 1). Highest benthic yields were in non-limited cultures inoculated with endolithic biomass ($p < 0.01$), suggesting that under these conditions filaments preferred to grow out of the substrate rather than deeper into it (see also Fig. 1a). These experiments suggest that strain BC008 preferentially fixes C from the most readily available source, switching when it runs low, and that DIC limitation enhances its propensity to initiate and remain boring.

**Mineral derived carbon is fixed in endolithic cells.** If strain BC008 were indeed fixing mineral-sourced carbon, the isotopic signature of the mineral should be detectable in its biomass. Figure 2a shows $\delta^{13}C$ values of endolithic, benthic and planktonic biomass fractions from long-term BC008 cultures, as well as those of the two potential sources. Mean $\delta^{13}C$ for each biomass type differed significantly ($p < 0.06$). Planktonic biomass fractionated its DIC source from $-4.2 \pm 0.5‰$ to a level typical of most planktonic autotrophs ($\delta^{13}C = -22.5 \pm 1.1‰$)[15]. If the sole C source for endolithic biomass had been DIC, one would have expected a $\delta^{13}C$ similar to that of planktonic biomass, or even slightly more negative, due to diffusive fractionation[16]. If the C source of endolithic biomass was mineral C, one would expect the $\delta^{13}C$ of endolithic biomass to be more positive (by ~ $5.6 \pm 1.2‰$) than that of planktonic biomass. Indeed, the difference ($8.8 \pm 2.4‰$), while slightly larger than expected, is consistent with that hypothesis. Benthic biomass gave an intermediate $\delta^{13}C$ (Fig. 2a) indicating that it may have incorporated carbon from both sources, or perhaps exchanged organic photosynthate from endolithic biomass while fixing DIC. To clarify the potential sources of C, particularly their relative contributions, we complemented our experiments with NanoSIMS imaging. For this, *Mastigocoleus* was incubated in closed cultures with $^{13}C$ enriched

(5479‰) DIC to follow the incorporation of this tracer vs. that of the mineral source (1.4‰). Control cultures incubated with tracer but no calcite substrate expectedly yielded very heavy biomass ($-4089 \pm 6.7‰$).

Qualitative observations show clear differences in cellular $\delta^{13}C$ as a function of the position of the cells in the boring bed (Fig. 2b). A quantitative survey across the boring bed (Fig. 2c) conspicuously shows a steep decrease in biomass $\delta^{13}C$ with depth. Cells just outside the solid surface showed some tracer signal, suggestive of some 12.5% of their C originating in DIC, and endolithic cells close to the surface were at around 5% of control biomass C, but the DIC influence faded quickly with depth, so that just 15 μm below the surface (a few cells deep), $\delta^{13}C$ become indistinguishable from that of the mineral (or that of control endoliths grown without tracer). Most endolithic biomass below the very surficial layer had no measurable contribution from external DIC. Hence, virtually all endolithic biomass C originated from mineral C.

**Natural endolithic communities also fix mineral carbon.** If laboratory results were generalizable to natural settings, one could predict that the $\delta^{13}C$ of euendolithic microbial communities may align closer to that of their mineral substrate than to that of surrounding seawater DIC. To test this we analyzed endolithic biomass and mineral substrates in a variety of samples collected around the intertidal of Isla de Mona, Puerto Rico (Fig. 3a), selecting sites showing discordant $\delta^{13}C$ values. Due to biological fractionation, biomass $\delta^{13}C$ was always more negative than any potential source. It was, however, not invariant, but rather a direct and linear function ($R^2 = 0.88$) of the substrate $\delta^{13}C$. This is consistent with the notion that organic C (as in our model system) stems from the lithospheric pool and reflects its mineral isotopic variability. These are complex communities[17], which often show fractionation intensities less than those of pure cultures[18]. In our case, they were around $9.58 \pm 1.93‰$, and were also a weak direct function of substrate $\delta^{13}C$ ($\Delta\delta^{13}C$ vs. $\delta^{13}C$, $R^2 = 0.62$), such that limestone endoliths, already quite depleted

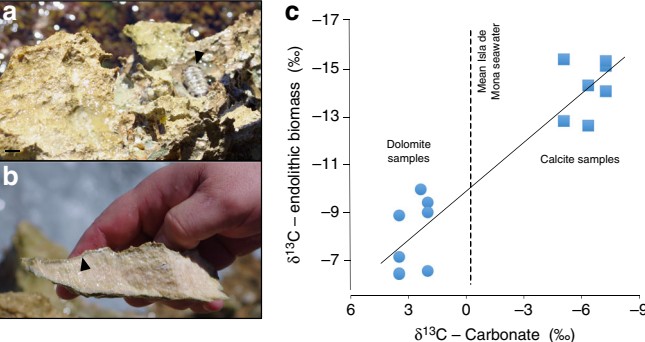

**Fig. 3** Carbonate substrates containing natural endolithic communities and $\delta^{13}C$ comparisons of endolith biomass against bored substrate. **a** Shows a typical carbonate sample substrate including a local mollusk grazer (arrowhead, *Chiton* sp. Polyplacophora). Scale bar is 2 cm. **b** Shows a manually cracked cross section of carbonate substrate, note the boring bed just under the rock surface (arrowhead). The carbonate substrates containing natural endolithic communities were located at six discreet intertidal sites on Isla de Mona, Puerto Rico. **c** Shows the differential $\delta^{13}C$ of organic endolithic biomass as a function of their respective substrate $\delta^{13}C$. The average mineral $\delta^{13}C$ for our limestone samples was $-6.2 \pm 1.1\permil$, whereas dolomite-rich samples were more positive ($2.6 \pm 0.8\permil$); all were quite apart from local seawater DIC ($-0.1\permil$), depicted by the vertical dotted line ($n = 3$, s.d. $= 0.005\permil$). A single-large infested rock fragment was coarsely cracked and three infested fragments of that were analyzed as site replicates for all but one dolomite site (very low infestation). Replicates are plotted vertically in a single column for each site

in $^{13}C$, had a significantly weaker fractionation than those in dolostone ($8.31 \pm 1.26$ vs. $11.20 \pm 1.23\permil$; ANOVA $p < 0.01$). Thus, field results are consistent with our culture experiments, and with the hypothesis that the main C source for endolithic communities is the local carbonate.

We demonstrated that euendolithic communities of cyanobacteria fix preferentially C liberated during mineral excavation, and that under DIC limitation, euendoliths prefer endolithic over benthic or planktonic growth. One could speculate that an evolutionary advantage of boring may have come about as a means to circumvent DIC limitation in benthic microenvironments. Given the old history of cyanobacterial euendoliths in the fossil record[19], this may have been an ancient adaptation. Moreover, because endoliths are actively consumed by grazers (Fig. 3a) and dissolution of solid carbonates will yield more C (about twice as much, Supplementary Table 1) than needed for biomass, it is likely that the isotopic signature of the mineral substrate will carry through the trophic web around ecosystems that contain endolithic communities, something that should be taken into consideration in (paleo) environmental $\delta^{13}C$ studies.

The surface extent of global carbonate outcrops is large, some $2 \times 10^7 \, km^2$[20], as is that of coral reefs ($10^5 \, km^2$)[21]. While detailed surveys of global endolithic biomass and productivity have yet to be carried out, the rates of phototrophic endolithic community productivity, at its likely optimum within tropical shallow carbonates, are quite large, upward of $2 \times 10^6 \, g \, C \, km^{-2} \, d^{-1}$[22]. One can therefore expect the global contributions of mineral-source autotrophy to be maximally in the $10^{10}$ tons of C annually, if likely only a fraction of that, which would, in any case, be globally relevant compared with the net global primary production estimated at some $100 \times 10^9$ tons of C[23].

## Methods

**Field samples**. Marine coastal endolithic communities were sampled from Isla de Mona (18.0867° N, 67.8894° W), a small, predominantly carbonate (Mona Dolomite and Lirio Limestone) island located 40 miles West of Puerto Rico[24]. Samples

were obtained from six separate localities along the shore, collected within the bioerosional notch of the intertidal zone. Rock samples containing endolithic biomass, verified post hoc using a digital field microscope, were broken from large boulders and rock walls using a standard geological hammer. Samples were allowed to dry and then shipped to the laboratory.

**Cultivation**. Cultures of the euendolith *Mastigocoleus testarum* strain BC008 were grown in standard, vented-cap, tissue culture flasks containing 15 ml of filter (0.22 μm) sterilized enriched seawater medium[25] medium. Briefly, the medium composition is as follows: $KNO_3$, 500 nM; $K_2HPO_4$, 50 pM; neutralized sodium silicate, 500 μM; 1.0 ml/l of a trace element solution containing per liter, 1.0 g $FeCl_3$. 6 $H_2O$, 620 mg $MnSO_4$. $H_2O$, 250 mg $ZnSO_4$.7 $H_2O$, 130 mg $Na_2MoO_4$.$2H_2O$, 4 mg $CoC1_2$.6 $H_2O$, 4 mg $CuSO_45H_2O$, and 6.0 g of disodium EDTA (sea water was obtained off the coast of Puerto Peñasco, Mexico, and filtered through a 0.22 μm filter. Vitamin stock solution was added to 2× final concentration). Cultures were incubated on a slowly rocking platform illuminated with a 16 h light/8 h dark diel cycle, at a light intensity of 22 μmol photon $m^{-2} \, s^{-1}$ from white fluorescent tubes. Boring cultures were obtained by adding ethanol-sterilized calcite chips (obtained from commercial blocky calcite, Ward's Scientific) of small size ( < 150 $mm^2$ surface area) to mineral-free liquid cultures, until the chips were colonized. Cultures used for stable isotope analysis were incubated for an average period of 6 months. The Saturation Index for calcite in our media (based on seawater composition and taking final pH into account) was 0.83. In closed incubations (see below), the SI was much higher (around 1.4) because of significant increases in pH.

**Dissolved $C_i$ limitation culture experiments**. Calcite chip fragments used in these experiments were chosen so as to provide equivalent surface area for colonization ($141 \pm 4 \, mm^2$).We used two types of biomass as inoculum: mineral-free biomass (or "planktonic") and biomass grown inside of calcite chips (or "endolithic"). Planktonic inoculum was obtained from cultures containing actively bored chips by harvesting biomass not associated with the mineral, which was re-suspended in fresh medium. The inoculum was homogenized by repeated passage through a 24 gauge sterile needle attached to a 3 ml syringe. Seed chips used for boring inoculum were obtained by incubating size-selected chips, without shaking, in $35 \times 10 \, mm$ tissue culture plates, sealed with Parafilm™, containing 3 ml of homogenized inoculum (obtained as explained above). Initiation of chip infestation was then monitored by optical microscopy, and as soon as radial boring colonies (Supplementary Fig. 1) were evident (about 3 weeks), chips containing between 15 and 20 radial colonies were collected, lightly brushed with sterile toothbrushes in sterile media and used as boring inoculum for the experiments.Triplicate independent cultures were grown for each type of inoculum (endolithic and planktonic), and under each of two conditions: open or closed to the atmosphere, so as to achieve differential $C_i$ limitation. All cultures were set up in 150 ml borosilicate glass serum bottles containing 50 ml of medium with an initial gas headspace of 100 ml. Inoculum consisted of either 100 μl of the planktonic homogenate together with three size-selected sterile naïve chips, or three seed chips. Cultures open to the atmosphere were not sealed but capped with sterile cotton-packed gauze. Closed cultures were hermetically sealed with aluminum and Teflon crimp caps. Headspace in sealed bottles was flushed with degassed Milli-Q™ water. Sterile, degassed Milli-Q™ water was added to open cultures, as needed, to keep volume at 50 ml, and make up for evaporative loss. Cultures were incubated for 4 months. Upon completion, planktonic biomass was collected from each bottle, rinsed three times in degassed Milli-Q™ water, pelleted, and dried at 60 °C for 72 h. Chips were removed from the culture vessel and gently rinsed with degassed sterile Milli-Q™ water in petri dishes, then brushed lightly with sterile toothbrushes to remove and harvest attached (but not endolithic) biomass (i.e., "benthic" biomass). Calcite chips were then placed in a 15 ml falcon tube and dissolved with 5 ml 1 N HCl to liberate endolithic biomass. After dissolution, tubes were spun, the supernatant discarded and the pelleted biomass rinsed with degassed sterile Milli-Q water then pelleted again. Endolithic biomass was then dried as described above. All dry biomass yields were determined gravimetrically. To gauge the depth of penetration in euendolithic growth, one chip from each culture was cracked into segments and the maximal penetration depth of the visible boring bed on the freshly opened faced was recorded using the stage micrometer of a stereomicroscope, with measurements on six surfaces ($n = 6$).

**Stable isotope analysis of biomass carbon**. Organic C and its isotopic composition were measured by continuous flow isotope ratio mass spectrometry (CF-IRMS) in biomass samples from either field or cultures, after carbonate dissolution in acid as explained above, using a Costech Elemental Analyzer, Thermo Conflo III, and Thermo Delta plus Advantage mass spectrometers. Biomass samples were encased in analytical tin capsules, combusted at 1020 °C in a reactor packed with chromium oxide and silvered cobaltous oxide, with flash combustion occurred on injection of a pulse of oxygen at the time of sample drop. Under these conditions, the tin capsule ignites, raising the sample temperature to 1800 °C. Combustion products were sent into a copper reduction reactor, where incomplete combustion products (NOx, CO, etc.) were reduced and excess $O_2$ is removed. Water is then chemically scrubbed from the helium. The final product gas ($CO_2$) was separated on a 3-meter Porapak-Q packed gas chromatography column. Sample analysis was interspersed with replicates of three different laboratory standards to ensure

instrument precision. These laboratory standards had been previously calibrated against NIST Standard Reference Materials (USGS40 and USGS41). The long-term standard deviation was 0.2‰ for $^{13}C$. All final $\delta^{13}C/^{12}C$ values are expressed in permil (‰), calculated using Eq. 1, where values are expressed relative to the international standard V-PDB (Vienna PeeDee Belemnite). All permil error is expressed as sample s.d.

$$^{13}\delta C_{PDB}(‰) = 1000((R_x/R_{PDB}) - 1) \quad (1)$$

where $R_x$ is the ratio of $^{13}C/^{12}C$ of the sample and $R_{PDB}$ is the ratio of $^{13}C/^{12}C$ ratio of PeeDee Belemnite (0.0112372).

**Stable isotope analysis of inorganic carbon.** The $\delta^{13}C$ content of all inorganic carbon (mineral and dissolved inorganic carbon) was determined after conversion to $CO_2$ through acidification by addition of excess phosphoric acid in Helium-flushed hermetically sealed vials, and incubation at 60 °C in a multiprep heating block. All sample inorganic carbon ends up in the headspace, where the $CO_2$ $\delta^{13}C$ is equivalent to that of the carbonate or dissolved $C_i$. $CO_2$ from the headspace was sampled using a Thermo GasBench, and $\delta^{13}C$ values obtained in an interfaced MAT 253 mass spectrometer

**NanoSIMS imaging experiments.** Two small ($3 \times 2 \times 2$ mm) calcite chips where incubated for one month with planktonic biomass to initiate light infestation (< 5% of the surface area infested), then collected, brushed with sterile toothbrushes, and rinsed in sterile media two times, to remove "benthic" outgrowth, and then incubated again for 5 days to allow broken filaments to self-repair. One chip was then placed in a 5 ml hermetically sealed vial containing 4.5 ml of $^{13}C$-enriched seawater medium (5479‰) and the other in another vial containing non-enriched medium. Two sets of two cultures were concurrently incubated as controls: the first set contained no inoculum, one vial with a chip and one vial without, the second set contained a planktonic inoculum with no chips, one in enriched medium and the other in non-enriched medium. Cultures were incubated under standard conditions for 31 days, after which headspace gas and liquid medium aliquots were analyzed by isotope ratio mass spectroscopy (IRMS) as described above. In addition, each chip (or planktonic biomass in the case of the control cultures) was removed, rinsed twice in degassed sterile Milli-Q™ water, then twice in sterile medium. The biomass in the chips at the end of the incubation typically covered some 90% of the surface at a much higher population density than initially (around 3.4%). Because of this we calculate that the probability of having imaged a cell that was present in the inoculum before the addition of tracer was < 1/1000, and that the probability of encountering four such cells successively at random (as we replicated our measurements) was less than one in one-hundred million. To ensure purging of all unfixed, heavy dissolved $C_i$, prior to further processing, all cultures were further incubated in regular, non-enriched medium for an additional four days, including a medium change on the second day. Chips were collected, rinsed with sterile medium, then with degassed sterile Milli-Q™ water and subjected to an acetone dehydration series (15, 25, 50, 75, and 100%), then completely embedded in Spurr's resin[26] stepwise (10, 25, 75, 100%), and polymerized under vacuum at 61 °C for 72 h. A single side of an embedded chip was exposed using a razor blade, and the calcite dissolved by 5 N HCl treatment. The resulting void space was then backfilled, under vacuum, with Spurr's resin and polymerized as before. Blocks were then trimmed to reveal the boring beds, which were then cut with a diamond knife in cross section for NanoSIMS imaging analysis.

Cut blocks were mounted in a single, custom made, 1-inch aluminum SIMS holder and coated with platinum using a Technics sputter coater. Control planktonic biomass was rinsed three times in degassed sterile Milli-Q™ water to remove salt and then deposited on a silicon wafer and dried at 60 °C for 72 h. The block surfaces containing the boring bed biomass embedded in resin were mapped isotopically with a NanoSIMS 50L. The position of the filaments was determined with the secondary electron image and the $^{12}C^{14}N^-$ ion image. The $^{12}C^{14}N^-$ ion image, owing to higher count rates, defines cell morphology better, sufficient to distinguish a clear boundary between cell wall and extracellular sheath of the cyanobacteria, which the $^{12}C^-$ image alone failed to achieve. A $Cs^+$ ion beam of 1–2 pA with a ~50 nm probe diameter was used to sputter the sample surface while secondary ions of $^{12}C^-$, $^{13}C^-$, and $^{12}C^{14}N^-$ were collected simultaneously using electron multipliers. Close isobaric interference by $^{12}C^1H$, $^{13}C$, $^{12}C^{14}N$, and $^{13}C_2$ were completely separated by using a high mass resolution (Cameca MRP > 8000). Before collecting the secondary ions, the sample surface was bombarded at high primary beam currents (200 pA) for about 10 min to achieve a steady-state secondary ion yield and remove any surface contamination. Images were acquired at a resolution of $256 \times 256$ pixels, while rastering the beam over an area of $15 \times 15$ μm or $20 \times 20$ μm, with a dwell time of 4 ms per pixel. The entire imaging sequence was performed by moving the stage from the outer part to the inner part of the original chip at multiple chip locations, thus collecting a depth profile through the boring bed.

Carbon isotope ratios were determined from $^{12}C^-$ and $^{13}C^-$ counts recorded, with corrections for electron-multiplier dead-time of 44 ns. The ion images were corrected for image drift and the $^{13}C/^{12}C$ ratios extracted by drawing regions of interest around each visible cell carefully excluding the extracellular sheath layer

using WinImage. No corrections due to Quasi Simultaneous Arrival[27] (< 10‰) were done given the magnitude of the enrichments in the labeled samples (up to 1000‰) and the internal analytical precision (~ 50‰) typical for these measurements. A control culture grown in non-enriched medium was identically embedded in resin and subsequently measured, and used to bracket between enriched sample measurements. The $^{13}C/^{12}C$ ratios of labeled cells were normalized to background $\delta^{13}C$ values obtained by measuring the embedding Spurr's resin. The $^{13}C/^{12}C$ ratios were then scaled to the Vienna PeeDee Belemnite standard (Eq. 1). The $\delta^{13}C$ uncertainties finally shown include the 1σ random Poisson component and second component due to the scatter obtained while doing reproducibility measurements on unlabeled cells.

**Statistics.** To test significant differences of biomass fraction yields between differing dissolved $C_i$ limitation treatments, multivariate analysis of variation (ANOVA) was used. Post-hoc comparisons among dissolved $C_i$ treatment and inoculation schemes was done by means of Tukey's HSD for each dissolved $C_i$ and inoculum level. One-way ANOVA analysis was ran for the comparisons of biomass fraction isotopic content. Statistical analyses were performed using IBM SPSS Statistics version 21 software.

**Data availability.** All data are available upon request from the authors in digital format.

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

## Acknowledgements

We would like to thank S. Romaniello and N. Zolotova for assistance with the gas bench and EA-IRMS analyses and P. Williams for usage of the NanoSIMS.

## Author contributions

B.S.G. and F.G.-P. designed the approaches and wrote the manuscript. B.S.G. carried out the experiments and analyses with M.B. aiding with nano SIMS. All authors edited the manuscript.

This work was primarily supported by NSF grant EAR-1224939, "Intra-cellular metal pumping in microbial excavation by microbes" from the LT Geochemistry and Geobiology Program, and by grant (EAR-1352996) for instrument support.

## Additional information

**Competing interests:** The authors declare no competing financial interests.

