## [Peer Review File · Nature Communications]

Reviewers' comments:

Reviewer #1 (Remarks to the Author):

Review of manuscript NCOMMS-16-25681 "Carbon fixation from mineral carbonates" by Prof Garcia-Pichel and colleagues.

The manuscript entitled "Carbon fixation from mineral carbonates" investigates the possibility of solid carbonates to serve as a carbon source for the euendolithic cyanobacterium *Mastigocoleus testarum* strain BC008. The authors employed growth experiments to study the preferred growth form under DIC limiting and non-limiting conditions, and further analyzed the stable isotope composition of endolithic, benthic and planktonic grown cultures. Based on these data in addition to growth experiments in ¹³C-labeled medium, the authors concluded that this euendolithic cyanobacterium can assimilate solid carbonates. They further claim that also natural endolithic communities at intertidal marine limestone and dolostone outcrops are capable of mineral-sourced autotrophy based on stable isotope composition measurements.

This is to my knowledge the first study presenting data that support the incorporation of carbon from lithospheric solid carbonates into microbial biomass. However, I have several comments regarding the data presentation in this study.

Summary paragraph:

Line 18: Regarding the statement "...cultures of euendolithic cyanobacteria"; since in this study only one cyanobacterium strain was investigated I would change this sentence to "cultures of an euendolithic cyanobacterium".

Line 18: Regarding the statement "derive the majority of their biomass carbon"; It is difficult to follow this conclusion without details on the calculation in the methods part of the manuscript. I assume this statement originates from the dilution of the ¹³C-label in the experiment with ¹³C-labeled medium (data shown in Figure 2 B/C).

Main text:

Line 101-102: As mentioned above, please provide the calculations in the Materials and Methods section based on which you estimated 13% and 6% of the biomass C originating from DIC.

Line 139/140: "...in the 10⁹ tons of C annually, which would account for a substantial proportion of the 100 x 10⁹ tons of C of global primary production." If I understand this correctly, mineral-source autotrophy would then contribute approximately 1% to global primary production? It might be a matter of interpretation, but I would not consider that a substantial contribution.

Figure 1E and line 361 in Supplementary Materials ("chips containing between 15-20 radial colonies were collected, lightly brushed with sterile toothbrushes in sterile media and used as boring inoculum for the experiments"): With the inoculum being between 15-20 radial colonies, are the authors sure that this is sufficiently exact to compare the amount of biomass developed in the growth experiment under non-limiting or DIC-limiting conditions?

Figure 2A:

It should be noted in the figure legend that IMR stands for the growth medium. Please also mention in the legend how these data were retrieved (in contrast to the NanoSIMS data in B and C).

Figure 2B:

Instead of calling it "high ratio" and "low ratio" on the legend, please provide exact values (at least minimum and maximum).

Are the left panels truly ¹²C images, or do they refer to ¹²C¹⁴N images as mentioned in the Supplementary Material for an improved definition of cell morphology (lines 453/454)?

In lines 468/469 in the Supplementary Material it is mentioned how ROIs were defined for the analysis. However, since there is quite some heterogeneity in the cell structure, please add the visual information on how the ROIs were defined (an example image showing the defined ROI). Lines 471-473: "A control culture grown in non-enriched medium was identically measured, and used to bracket between enriched sample measurements." However, I would think a better suited control would have been cells grown in non-enriched medium, but then also embedded in Spurr's resin, the same way the enriched samples were treated for analysis. Spurr's resin does contain carbon so that the isotopic signature of the cells could have been altered due to infiltration by the resin. To take this into consideration, the control should have been treated the same way. Lines 278-279: "B and C correspond to cultures incubated with ¹³C-DIC enriched medium, and started with very lightly bored inocula (Fig. S2), so that the virtual totality of the biomass grew in the presence of the tracer." In order to demonstrate that the analyzed cells were truly newly synthesized biomass, can the authors provide additional images demonstrating in what area of the section the NanoSIMS images were taken, and that at the point of inoculum no biomass was present in those regions? This would clearly demonstrate that the cells with $\delta^{13}\text{C}$ values close to the control (the cells deepest in the solid) do not simply show this value because they were not dividing in the experiment.

Figure 3:

Line 302: I would suggest "Stable isotope composition analyses of biomass..." rather than "Stable C isotope analyses of biomass..."

It is a bit unclear how many sites and replicates of each site were measured. Please clarify 1) in line 302/303 how many sites per stone type were sampled, and b) in line 309 that of one sample only one "replicate" was analyzed. Also in Figure C, please specify which are the replicates per site.

Reviewer #2 (Remarks to the Author):

In this study the authors perform lab experiment using the cyanobacterial strain *M. testarum* to colonize calcite crystals. They show that the dissolution of calcite induced by *M. testarum* can compensate a limitation of the solution in bicarbonates. By NanoSIMS they confirm that C in calcite is incorporated in the cells. Then they try to expand this to some field cases.

There is some potential confusion that may rise from the title and the introduction dealing with C cycle, as if solid C was directly used/transferred to the cells. The authors did not evidence direct electron transfer to the solid such as those occurring in other instances (on solid Fe-oxides by DIRB). Of course, one understand from the study that C transfer from calcite to the cells occurs through calcite dissolution. Such an exchange of C between solid carbonates, DIC and atmospheric C always happen even in the absence of endolithic cyanobacteria. From what I understand here, the authors mean that cyanobacteria accelerate the exchange from carbonate to DIC then biosphere by dissolving carbonates but this was already shown by them in the past. However what I find very interesting is the fact that the dissolution of carbonates may prevent DIC limitation for these bacteria and the tracing by isotopic labelling of the C from the carbonate to the cells. However, I have several comments on this manuscript that would need to be addressed to make it clearer:

There is really a mystery here about the fate of Ca. If C contained by CaCO_3 goes to organic matter (CH_2O) then Ca^{2+} and hydroxyls should be released and I guess with CO_2 that might induce CaCO_3 precipitation. Here a suggestion of a balanced chemical equation should be provided (e.g., $\text{CaCO}_3 + \text{H}_2\text{O} + \text{CO}_2 = \text{CH}_2\text{O} + \text{O}_2 + \text{CaCO}_3$ Instead of $\text{Ca}^{2+} + 2\text{HCO}_3^- = \text{CH}_2\text{O} + \text{O}_2 + \text{CaCO}_3$). Would that mean that locally around the cells pH rises but DIC decreases? Or is this buffered by CaCO_3 precipitation?

Another problem is that we have no clue about the saturation index in these experiments. This is really important if one wants to understand what is expected thermodynamically. Moreover, is the ability to dig is connected to the distance to saturation of the solution? If DIC decreases, the saturation decreases and therefore thermodynamically speaking it may be easier to dissolve carbonates?

Isotopes: What are the initial compositions? If I add water to calcite, there will be calcite dissolution. At equilibrium, what is the expected isotopic composition of C in DIC?

How can you be sure that you got rid of all carbonates associated with endoliths? Acid dissolves carbonates but at some point carbonate dissolution buffers acidity. If there was some C from carbonates left, then this would have a strong impact on the isotopic composition measurements of endoliths.

- Fig. 2B: images 12C: What do we see? I see a cell on the outside, at the surface but not in the deep part (at least not at the same scale). What is the scale for the 12C map? Why not the same color code?

- Table S1: could you give some kind of mass balance that would support that? How much of the chip needs to be dissolved? Does it correspond to the endolith volume?

- Biomass in endoliths in labelled experiments: is it biomass that was produced during the experiment? Maybe it was produced before the experiment, hence not surprising that you do not see enrichment

- L74: "bore deeper": How much deeper? Statistically robust?

- "If laboratory results were generalizable to natural settings, one could predict that the $\delta^{13}\text{C}$ of euendolithic microbial communities may align closer to that of their mineral substrate than to that of surrounding seawater DIC": But I guess the carbonates they dig in formed in the seawater they live in and hence there should not be any difference isotopically between HCO_3^- release by the carbonates and HCO_3^- in the water. Moreover, I expect huge temporal variations of the $\delta^{13}\text{C}$ of the DIC in intertidal systems, shouldn't I?

Reviewer #3 (Remarks to the Author):

Carbon fixation from mineral carbonates
By Guida BS, Bosse M, Garcia-Pichel F.

The contribution documented lithospheric carbonate as a source of photosynthetic carbon fixation by microboring cyanobacteria. The results are novel and of interest of the entire scientific community as well as public.

This is a fascinating paper with important results and conclusions, however the reading is uninspiring and it needs a re-assessment. The paragraphs are too bulky, they need to be broken down so that the new divisions are building up the case convincingly. With full understanding of the space limitation directed by the style of the Journal (130 lines of main text – 500 lines in supplement text), the reader should not miss the essential information from the Materials and Methods that usually precedes the text. The needed information should be inserted in a compact form in appropriate places in the main text (see lines 64 and 388).

There are some significant differences in the data obtained, which were not explained, but should at least be addressed and/or discussed: For example, the Fig. 3C shows some significant

differences in the results. The experimental setting involves two inoculum sources ("planktonic" and endolithic), two conditions (DiC-unlimited and DiC-limited) and three growth stages where the responses were recorded (endolithic, epilithic and "planktonic"). In addition to significant differences between respective C+ and C- conditions that support the hypothesis forwarded by the authors, other also significant differences were recorded between the results of the two sets of exposures departing from the two types of inocula, "planktonic" and endolithic (that originated from the same cultured strain), both when compared within the C-unlimited or within the C-limited exposures. Why are these responses so different? –

Recommendation, moderate modification.

Minor corrections and comments by lines:

14 – Insert: ', the other potential carbon pool is the litho-hydrospheric carbonate.

41 – 'extrusion' – specify the form of extrusion: is it extruded as dissolved or as precipitated carbonate, or both?

46-49 – a cumbersome run-on sentence try the following instead:

46 – their own habitat is attractive for the following reasons: It completes the geo-microbial action on the substrate, and provides the euendoliths with a competitive advantage over photosynthetic epiliths in case of C (DIC) limitation, which may be caused by biofilms thickening 11, 12

50 – change: 'principally' to read: ', in principle,'

55-79 – This paragraph needs to be broken down into smaller blocks so as the content changes.

56 – please explain why is "planktonic" under quotation marks (because it is not really planktonic?) and what does this term mean in the context of the present study. See also line 62, 68. In other places it is used without quotation marks (e.g. Line 258) but it is compared with the results from the real plankton. I would prefer a more neutral terminology, e.g. suspended as opposed to settled or epilithic as opposed to endolithic. See also line 84.

63-64 – unclear, please rephrase. Replace: in its absence to read: without penetrating it (unbored).

64-65 – Here one is missing the Materials and Methods to see whether the variation is standard deviation or standard error. Put short explanation such as (mean \pm St.Dev. or St.Er.) where first mentioned – See line 388 below!

71 – Break the paragraph at: Expectedly, but start the sentence with: Highest yields

85 – Here you mean the real plankton. You need to explain the similarities and/or differences.

97 – Break the paragraph starting with: Qualitative observations...;

130 – Table 1S should be Table 1 in the main text. See also Comments to Lines 515-538

Fig. 1

255 – insert "both" for clarity to read: (scale bars = 0.5 mm) both show calcite chips with...

259 – The color intensities in the Fig. 2 are confusing. Apparently, A goes with C, and B goes with D. which is not helped by the darkness of the colors nor by the sequence of explanations in the caption. To make it clearer, I suggest to mark the thickness of the epilithic biofilm with a line in

both A and B rather than with arrowheads in A only. E – the diagram shows significant differences that need explanation See the introduction of this review.

260 – Change: Panel F shows biomass yield distributions (average-e) among growth mode, according to carbon limitation treatment. To read: Relative distribution of the biomass yield by growth mode under DIC-unlimited and -limited growth conditions.

Figure 2

Figure 3

302 – Change the first sentence identifying the Figure 3 to read: Carbonate substrates with natural endolithic communities (located at six discreet intertidal sites) on Isla de Mona, PR. and stable C isotopes in endolith biomass vs bored substrate.

I would prefer this to be Fig. 1 and that the article starts with the description of natural setting. The isotope distribution between dolomite and calcite compared with the corresponding endolith biomass is just as intriguing as the hypotheses the article has started with, and will better fit into the flow of the arguments and the experimental proofs delivered.

304 – change: (arrowhead) to read: (Chiton sp. Polyplacophora, arrowhead)

304-305 – The figure 3B demonstrates how to access chasmoendoliths rather than euendoliths, because the rocks crack preferentially along pre-existing fissures that are abundant in limestone. Did you try to scrape the green zone after brief etching? If it stays green, you have euendoliths. Please do that.

306 – Figure 3C legend is missing. Please add legend for the blue spheres and squares, or better yet, simply add the text DOLOMITE and LIMESTONE in the picture.

308 – The compact explanation for the variations in data is only here in the caption of Fig. 3 but it should be in the text, where the data first occur in the paper: see line 64.

Supplements

335 – change: boring notch to read: bored or bioerosional notch. (because the notch is not boring)

342 – It would be useful and desirable to briefly describe the composition of the medium used, since the reference is over 50 years old.

342, 425 – Please spell out what does IMR stand for (Internet lists 46 possibilities)

515-538 – The table 1 looks more like a Figure with an enormous caption, but it is an important contribution and should be in the main text. Suggestion: Transfer the Table in the main part of the contribution and incorporate its "caption" into the main text.

537 – Change: Chitons and mollusks. To read: Chiton and other mollusks.

Respectfully submitted,
Stjepko Golubic

Our Responses to the reviewers' comments:

All edited text can be seen in blue font in our manuscript file.

Reviewer #1 (Remarks to the Author):

Review of manuscript NCOMMS-16-25681 "Carbon fixation from mineral carbonates" by Prof Garcia-Pichel and colleagues.

The manuscript entitled "Carbon fixation from mineral carbonates" investigates the possibility of solid carbonates to serve as a carbon source for the euendolithic cyanobacterium *Mastigocoleus testarum* strain BC008. The authors employed growth experiments to study the preferred grown form under DIC limiting and non-limiting conditions, and further analyzed the stable isotope composition of endolithic, benthic and planktonic grown cultures. Based on these data in addition to growth experiments in ^{13}C -labeled medium, the authors concluded that this euendolithic cyanobacterium can assimilate solid carbonates. They further claim that also natural endolithic communities at intertidal marine limestone and dolostone outcrops are capable of mineral-sourced autotrophy based on stable isotope composition measurements.

This is to my knowledge the first study presenting data that support the incorporation of carbon from lithospheric solid carbonates into microbial biomass. However, I have several comments regarding the data presentation in is study.

Summary paragraph:

Line 18: Regarding the statement "...cultures of euendolithic cyanobacteria"; since in this study only one cyanobacterium strain was investigated I would change this sentence to "cultures of an euendolithic cyanobacterium".

Thank you, fixed.

Line 18: Regarding the statement "derive the majority of their biomass carbon"; It is difficult to follow this conclusion without details on the calculation in the methods part of the manuscript. I assume this statement originates from the dilution of the ^{13}C -label in the experiment with ^{13}C -labeled medium (data shown in Figure 2 B/C).

Thank you for pointing that out. The reviewer is correct, in that the evidence for the claim stems from Fig. 2B/C. The fact that the signal matches overwhelmingly the signature of the carbonate, rather than that of the externally supplied tracer, even without an explicit calculation, clearly demonstrates this fact. The data are explicit in the body of the paper, as a major finding from our analysis. In that sense we do not think it is necessary to also explicitly address it in the abstract.

Main text:

Line 101-102: As mentioned above, please provide the calculations in the Materials and Methods section based on which you estimated 13% and 6% of the biomass C originating from DIC.

The numbers are simply a percentage of the control ^{13}C signal. As this experiment was done in a closed system, inoculated with an insignificant amount of starting biomass and having only 2 sources of carbon, the DIC (all heavy) and the chip (all light, relatively) we can infer (again relatively) assuming that the

control (no chip, all heavy DIC) ^{13}C signal represents 100% ^{13}C incorporation. The only possible source of lighter C is the chip. For example, cells just outside the chip surface (-4.5 μm , Fig. 2-C) have an average $\delta^{13}\text{C}$ of circa 510‰, which is ~12.5% of the heavy control (Table S1, 4089‰). This is explicitly stated in the text.

Line 139/140: "...in the 109 tons of C annually, which would account for a substantial proportion of the 100×10^9 tons of C of global primary production." If I understand this correctly, mineral-source autotrophy would then contribute approximately 1% to global primary production? It might be a matter of interpretation, but I would not consider that a substantial contribution.

(Now line 135) An unknown source that adds 1% is substantial in our view. For comparison, in the global carbon cycle, researchers have been worrying for years about a "missing carbon sink" that accounts for some $1\text{-}2 \times 10^9$ tons of C, a magnitude similar to our contributions from endoliths.

And, please note that we did not want to abuse extrapolation of single points and left the statement quite vague, not actually saying that it was 1%. In fact, extrapolating the actual data from tropical reefs, one could easily come up with some 17% of the total. However, this productivity value belongs to probably the most productive endolithic systems of all, given that they enjoy high light, consistently high temperatures, and the continued presence of liquid water, which obviously does not apply to a large portion of carbonate outcrops. Given that the typical productivity in algal beds and reefs is just below one order of magnitude higher than global averages, we still feel that our soft statement probably depicts a better estimate. We have re-written the paragraph accordingly.

Figure 1E and line 361 in Supplementary Materials ("chips containing between 15-20 radial colonies were collected, lightly brushed with sterile toothbrushes in sterile media and used as boring inoculum for the experiments"): With the inoculum being between 15-20 radial colonies, are the authors sure that this is sufficiently exact to compare the amount of biomass developed in the growth experiment under non-limiting or DIC-limiting conditions?

(Now line 333) Yes we are. The generation time of this organism is very long (>96 hrs while endolithic) and unlike standard bacteria (*E. coli*) not all cells are in a replicative stage. Thus, its growth isn't the canonical exponential most microbiologists are accustomed to, especially when growing as an endolith. Radial colonies eventually fuse to form a single boring bed. Each experimental replicate contained several chips of approximate equal surface area, with roughly the same total number of visible radial colonies per culture replicate. The deviations in biomass obtained per culture replicate (yield) are quite small suggesting that the culture seeding was uniform enough among them to yield robust statistical comparisons.

Figure 2A:

It should be noted in the figure legend that IMR stands for the growth medium. Please also mention in the legend how these data were retrieved (in contrast to the NanoSIMS data in B and C).

Great suggestions, fixed.

Figure 2B:

Instead of calling it "high ratio" and "low ratio" on the legend, please provide exact values (at least minimum and maximum).

The minimum and maximum ratio values have been added to the figure legend.

Are the left panels truly ^{12}C images, or do they refer to $^{12}\text{C}^{14}\text{N}$ - images as mentioned in the Supplementary Material for an improved definition of cell morphology (lines 453/454)?

The images shown are indeed of the ^{12}C ion; the CN images were used to get an orientation and to assist in pinpointing sub-cellular features for ROI assignment. Some of the ^{12}C images (those towards the surface of the chip) were as equally resolved as the CN images.

In lines 468/469 in the Supplementary Material it is mentioned how ROIs were defined for the analysis. However, since there is quite some heterogeneity in the cell structure, please add the visual information on how the ROIs were defined (an example image showing the defined ROI).

We have added a supplementary figure (Fig. S3.) illustrating an example ROI delineation, also pointing out some landmark cellular features used in the assignment.

Lines 471-473: "A control culture grown in non-enriched medium was identically measured, and used to bracket between enriched sample measurements." However, I would think a better suited control would have been cells grown in non-enriched medium, but then also embedded in Spurr's resin, the same way the enriched samples were treated for analysis. Spurr's resin does contain carbon so that the isotopic signature of the cells could have been altered due to infiltration by the resin. To take this into consideration, the control should have been treated the same way.

(Now Line 446) Our apologies for not being clear here. The reviewer's suggestion is exactly what was done, the control culture was, in fact, a chip grown in non-enriched medium that was then embedded. This allowed us to take the resin into consideration when we normalized to generate accurate standard permil (vs. PDB) values. We have added a small line for further clarification.

Lines 278-279: "B and C correspond to cultures incubated with ^{13}C -DIC enriched medium, and started with very lightly bored inocula (Fig. S2), so that the virtual totality of the biomass grew in the presence of the tracer." In order to demonstrate that the analyzed cells were truly newly synthesized biomass, can the authors provide additional images demonstrating in what area of the section the NanoSIMS images were taken, and that at the point of inoculum no biomass was present in those regions? This would clearly demonstrate that the cells with $\delta^{13}\text{C}$ values close to the control (the cells deepest in the solid) do not simply show this value because they were not dividing in the experiment.

The sterile chips used were initially infested in heavy medium using a very small starting planktonic inoculum within a closed system. Thus all endolithic biomass must have originated using either heavy carbon from the medium or light carbon from the chip itself (or a combination of both). The probability that we imaged cells originating from the starting inocula, a month later, as endolithic biomass, is vanishingly small.

Figure 3:

Line 302: I would suggest "Stable isotope composition analyses of biomass..." rather than "Stable C isotope analyses of biomass...".

(Now line 286) Good suggestion; it was incorporated, see below.

“Carbonate substrates containing natural endolithic communities (located at six discreet intertidal sites) on Isla de Mona, PR. and $\delta^{13}\text{C}$ comparisons of respective endolith biomass vs bored substrate.”

It is a bit unclear how many sites and replicates of each site were measured. Please clarify 1) in line 302/303 how many sites per stone type were sampled, and b) in line 309 that of one sample only one “replicate” was analyzed. Also in Figure C, please specify which are the replicates per site.

We added a clarifying line to the figure caption that should prevent confusion, and indicated site types (calcite vs dolomite) on the graph itself.

Reviewer #2 (Remarks to the Author):

In this study the authors perform lab experiment using the cyanobacterial strain *M. testarum* to colonize calcite crystals. They show that the dissolution of calcite induced by *M. testarum* can compensate a limitation of the solution in bicarbonates. By NanoSIMS they confirm that C in calcite is incorporated in the cells. Then they try to expand this to some field cases.

There is some potential confusion that may rise from the title and the introduction dealing with C cycle, as if solid C was directly used/transferred to the cells. The authors did not evidence direct electron transfer to the solid such as those occurring in other instances (on solid Fe-oxides by DIRB). Of course, one understand from the study that C transfer from calcite to the cells occurs through calcite dissolution. Such an exchange of C between solid carbonates, DIC and atmospheric C always happen even in the absence of endolithic cyanobacteria. From what I understand here, the authors mean that cyanobacteria accelerate the exchange from carbonate to DIC then biosphere by dissolving carbonates but this was already shown by them in the past.

It has been shown that *M. testarum* can dissolve different carbonate substrates (Ramirez-Reinat and Garcia-Pichel, 2012) but the mechanism of dissolution has only recently been demonstrated and only quantitatively in reference to the cationic component of the solid. The fate of the anionic carbonate component had yet to be confirmed. The exact molecular mechanism of the carbonate reaction and where it takes place is also yet unaddressed; we do not claim that direct electron transfer to the carbonate is happening. In fact, we would doubt that it is the case, as it would not bring any additional fitness value to these organisms. We do not see the need of a direct electron transfer to the mineral as a requirement for our claim in the title. The mechanism involves an initial step of dissolution but the fate of that dissolved carbonate is for the exclusive use of the euendolith. This involves the exclusive agency of the model euendolith.

However what I find very interesting is the fact that the dissolution of carbonates may prevent DIC limitation for these bacteria and the tracing by isotopic labelling of the C from the carbonate to the cells.

Thanks, we did find that interesting as well.

However, I have several comments on this manuscript that would need to be addressed to make it clearer:

There is really a mystery here about the fate of Ca. If C contained by CaCO_3 goes to organic matter (CH_2O) then Ca^{2+} and hydroxyls should be released and I guess with CO_2 that might induce CaCO_3 precipitation. Here a suggestion of a balanced chemical equation should be provided (e.g., $\text{CaCO}_3 + \text{H}_2\text{O}$

+ CO₂=CH₂O+ O₂+CaCO₃ Instead of Ca²⁺+ 2HCO₃⁻ = CH₂O+O₂ + CaCO₃). Would that mean that locally around the cells pH rises but DIC decreases? Or is this buffered by CaCO₃ precipitation?

Yes, although the fate of calcium has been addressed elsewhere (Garcia-Pichel et al, 2010; Guida and Garcia-Pichel, 2016; see citations 8, 9 and 10). Active intracellular transport of Ca²⁺ and protons occurs in the long range, whereby Ca²⁺ gets excreted to the outside medium (and protons imported) at the expense of energy. Thus a complete balanced reaction cannot be assigned to the boring cells without including what happens well beyond the endolithic habitat. Indeed carbonate reprecipitation as micrite is known to occur away from the boring front in nature and in culture. But that effect, while relevant to the boring action, we see as being beyond the scope of this paper.

Another problem is that we have no clue about the saturation index (SI) in these experiments. This is really important if one wants to understand what is expected thermodynamically. Moreover, is the ability to dig is connected to the distance to saturation of the solution? If DIC decreases, the saturation decreases and therefore thermodynamically speaking it may be easier to dissolve carbonates?

That is correct. However, sea water (and our medium) *is* saturated relative to the mineral substrate (calcite). There is little doubt that the saturation index at the boring front is altered relative to outside but the degree of alteration at the spatial scales we'd need to measure (submicron) would be extremely difficult (if not impossible with current technology) to ascertain. Thus, it seems likely (thermodynamically anyway) that the initiation of boring from the outside of the chip would be much more difficult than simply maintaining the boring process at a lower SI inside the chip. This makes sense as the ATP energy cost of living inside the rock would then be lower as well. These we see as questions that have yet to be addressed and are also beyond the scope of this work.

Isotopes: What are the initial compositions? If I add water to calcite, there will be calcite dissolution. At equilibrium, what is the expected isotopic composition of C in DIC?

If you add pure water to calcite there will be slow dissolution until equilibrium is reached then you will see extremely slow isotopic exchange. We demonstrated such an isotopic exchange in our experiments in Table S1. On the time scale of this experiment, any isotopic exchanges between the liquid and solid phases were negligible. The initial isotopic compositions are listed in the text for our medium (-4.2‰), sea water (-0.5‰), and our calcite substrate (1.4‰).

How can you be sure that you got rid of all carbonates associated with endoliths? Acid dissolves carbonates but at some point carbonate dissolution buffers acidity. If there was some C from carbonates left, then this would have a strong impact on the isotopic composition measurements of endoliths.

Small calcite chips, once embedded in resin, were placed in 250mL of 5N HCL for 24 hours, after which no additional effervescence was observed under a dissecting microscope. Simple stoichiometric calculations show that the HCL is well in excess of the rather small calcite chips. However, it is possible that submicron sized calcite fragments (potentially shielded by embedded biomass) may have survived the process. Even then, we measured only inside visible cells (see new Fig S3) and some areas surrounding the cells we did measure for background normalization purposes, no spikes in ¹³C were observed confirming that no small extracellular calcite fragments survived the HCL treatment within the areas we analyzed.

- Fig. 2B: images 12C: What do we see? I see a cell on the outside, at the surface but not in the deep part (at least not at the same scale). What is the scale for the 12C map? Why not the same color code?

The sizes of individual cells as they bore changes dramatically, with the average diameter being 10 - 12 microns outside and 4 -5 microns in the boring bed. There are at least 5 individual cells with clearly delineated borders, and we have added arrows in Fig. 2 for clarification. Different areas of the chip have slightly different surface topology (such as pitch and pitting) thus the imaging parameters must be optimally adjusted as we analyze different chip areas.

- Table S1: could you give some kind of mass balance that would support that? How much of the chip needs to be dissolved? Does it correspond to the endolith volume?

This was a relatively short experiment generating very little biomass, which was not quantified nor did we quantify the amount of DIC left at the conclusion of the experiment. We were only interested in the potential release of stored mineral carbonate from actively boring biomass, and thus only looked at the isotopic dilution.

- Biomass in endoliths in labelled experiments: is it biomass that was produced during the experiment? Maybe it was produced before the experiment, hence not surprising that you do not see enrichment

All biomass produced here was under isotopically enriched closed conditions. See previous comment by reviewer 1.

- L74: "bore deeper": How much deeper? Statistically robust?

Yes, about 50 μm deeper ($p < 0.01$), please see supplementary Figure S1.

- "If laboratory results were generalizable to natural settings, one could predict that the $\delta^{13}\text{C}$ of euendolithic microbial communities may align closer to that of their mineral substrate than to that of surrounding seawater DIC": But I guess the carbonates they dig in formed in the seawater they live in and hence there should not be any difference isotopically between HCO_3^- release by the carbonates and HCO_3^- in the water. Moreover, I expect huge temporal variations of the delta ^{13}C of the DIC in intertidal systems, shouldn't I?

Based upon the mechanism at which different types of calcium carbonate minerals (such as calcite vs dolomite) crystalize and their content of organic carbon, carbonate minerals have a very different isotopic composition from their surrounding seawater (hence their use in paleosedimentology). We showed this for our sample sites on Isla de Mona (Fig. 3-C). While the exposed layer of any carbonate would, over long periods of time, isotopically equilibrate with the local seawater, the interior would maintain the mineral's isotopic signature at the time of its formation. Any carbon fixing euendoliths deep enough (about 15 microns according to our data) should reflect the mineral isotopic signature, which is exactly what we found. Few studies have followed $\delta^{13}\text{C}$ values of individual existing intertidal systems for any length of time but there are small annual fluctuations. We know that variation is dependent on many factors such as surface water temperature, evaporation rates, eutrophic water upwelling (thus photosynthetic activity) and atmospheric CO_2 input, these changes appear modest around $\pm 3 \text{ ‰}$ at any particular site studied (Paver, 2007 - Thesis and Williams et al. 2011.)

Temporal analysis of endolithic biomass delta ¹³C values would be a great corollary to this work and we predict that no significant variation would be seen based upon the results presented here.

Reviewer #3 (Remarks to the Author):

Carbon fixation from mineral carbonates

By Guida BS, Bose M, Garcia-Pichel F.

The contribution documented lithospheric carbonate as a source of photosynthetic carbon fixation by microboring cyanobacteria. The results are novel and of interest of the entire scientific community as well as public.

This is a fascinating paper with important results and conclusions, however the reading is uninspiring and it needs a re-assessment. The paragraphs are too bulky, they need to be broken down so that the new divisions are building up the case convincingly. With full understanding of the space limitation directed by the style of the Journal (130 lines of main text – 500 lines in supplement text), the reader should not miss the essential information from the Materials and Methods that usually precedes the text. The needed information should be inserted in a compact form in appropriate places in the main text (see lines 64 and 388).

There are some significant differences in the data obtained, which were not explained, but should at least be addressed and/or discussed: For example, the Fig. 3C shows some significant differences in the results. The experimental setting involves two inoculum sources (“planktonic” and endolithic), two conditions (DIC-unlimited and DIC-limited) and three growth stages where the responses were recorded (endolithic, epilithic and “planktonic”). In addition to significant differences between respective C+ and C- conditions that support the hypothesis forwarded by the authors, other also significant differences were recorded between the results of the two sets of exposures departing from the two types of inocula, “planktonic” and endolithic (that originated from the same cultured strain), both when compared within the C-unlimited or within the C-limited exposures. Why are these responses so different? –

Recommendation, moderate modification.

Minor corrections and comments by lines:

14 – Insert:’, the other potential carbon pool is the litho-hydrospheric carbonate.

We are not sure what the reviewer means here, carbonate is either in mineral form, or dissolved in solution (the exact species being pH dependent)

41 – ‘extrusion’ – specify the form of extrusion: is it extruded as dissolved or as precipitated carbonate, or both?

It is extruded in its ionic form. Clarification has been added.

46-49 – a cumbersome run-on sentence try the following instead:

46 – their own habitat is attractive for the following reasons: It completes the geo-microbial action on the substrate, and provides the euendoliths with a competitive advantage over photosynthetic epiliths in case of C (DIC) limitation, which may be caused by biofilms thickening 11, 12

Thank you! We have incorporated your suggestions and reworded the thought.

50 – change: ‘principally’ to read: ‘, in principle,’
Good catch, the sentence has been reworked.

55-79 – This paragraph needs to be broken down into smaller blocks so as the content changes.
Good suggestion, done.

56 – please explain why is “planktonic” under quotation marks (because it is not really planktonic?) and what does this term mean in the context of the present study. See also line 62, 68. In other places it is used without quotation marks (e.g. Line 258) but it is compared with the results from the real plankton. I would prefer a more neutral terminology, e.g. suspended as opposed to settled or epilithic as opposed to endolithic. See also line 84.

Planktonic, in our context, refers to suspension growth, similar to plankton, nowhere in our text are we truly referring to canonical plankton organisms, hence the initial quotes, which tell the reader the word will not be used in the standard context. We have added additional clarification after our initial use of the term in the text.

Line 55 - (producing “planktonic” or suspended biomass)

63-64 – unclear, please rephrase. Replace: in its absence to read: without penetrating it (unbored).

The subject of “it” here was the DIC so the line has been changed to “in the absence of DIC.”

64-65 – Here one is missing the Materials and Methods to see whether the variation is standard deviation or standard error. Put short explanation such as (mean \pm St.Dev. or St.Er.) where first mentioned – See line 388 below!

We added a clarifying line in the M&M stating all percent error is reported as standard deviation (line 406-407)

71 – Break the paragraph at: Expectedly, but start the sentence with: Highest yields

Thank you, changes made, it reads much better now.

85 – Here you mean the real plankton. You need to explain the similarities and/or differences.

No, we do not analyze oceanic plankton in this study. We added some further clarification here.

Lines 81 - 83, Figure 2A shows $\delta^{13}\text{C}$ values of endolithic, benthic and planktonic biomass fractions from long-term BC008 cultures, as well as those of the two potential sources.

97 – Break the paragraph starting with: Qualitative observations...;

Done.

130 – Table 1S should be Table 1 in the main text. See also Comments to Lines 515-538

Unfortunately the word length and figure limits of the journal will not permit us to do that. This was our original intention.

Fig. 1

255 – insert “both” for clarity to read: (scale bars = 0.5 mm) both show calcite chips with...

Thank you, done! (Now line 246)

259 – The color intensities in the Fig. 2 are confusing. Apparently, A goes with C, and B goes with D. which is not helped by the darkness of the colors nor by the sequence of explanations in the caption. To make it clearer, I suggest to mark the thickness of the epilithic biofilm with a line in both A and B rather than with arrowheads in A only. E – the diagram shows significant differences that need explanation See the introduction of this review.

(Now line 249) All differences were explained and discussed between lines 55 and 79 in the main text. This thickness of the epilithic biomass is very different at different positions around the chip so a line such as you suggested would be misleading.

260 – Change: Panel F shows biomass yield distributions (average-e) among growth mode, according to carbon limitation treatment. To read: Relative distribution of the biomass yield by growth mode under DIC-unlimited and -limited growth conditions.

Great suggestion, done! (Now line 252)

Figure 3

302 – Change the first sentence identifying the Figure 3 to read: Carbonate substrates with natural endolithic communities (located at six discreet intertidal sites) on Isla de Mona, PR. and stable C isotopes in endolith biomass vs bored substrate.

I would prefer this to be Fig. 1 and that the article starts with the description of natural setting. The isotope distribution between dolomite and calcite compared with the corresponding endolith biomass is just as intriguing as the hypotheses the article has started with, and will better fit into the flow of the arguments and the experimental proofs delivered.

(Now line 268) Analysis of the natural setting was an extension of the lab experiments with BC008 (much more controlled) and we would like to present the strongest evidence first, for example, we extrapolated those *in vitro* findings to the natural setting and not vice-versa.

304 – change: (arrowhead) to read: (Chiton sp. Polyplacophora, arrowhead)

(Now 270) Good find, we missed that, thank you!

304-305 – The figure 3B demonstrates how to access chasmoendoliths rather than euendoliths, because the rocks crack preferentially along pre-existing fissures that are abundant in limestone. Did you try to scrape the green zone after brief etching? If it stays green, you have euendoliths. Please do that.

(Now line 271) These are definitely endoliths. The rock is not limestone but was very hard and non-porous. All samples were scrubbed with a wire brush for confirmation.

306 – Figure 3C legend is missing. Please add legend for the blue spheres and squares, or better yet, simply add the text DOLOMITE and LIMESTONE in the picture.

Thanks, fixed.

308 – The compact explanation for the variations in data is only here in the caption of Fig. 3 but it should be in the text, where the data first occur in the paper: see line 64.

The explanations for these data are located between lines 80 and 97.

Supplements

335 – Change: boring notch to read: bored or bioerosional notch. (Because the notch is not boring)

(Now line 302) You are correct, thanks. Fixed

342 – It would be useful and desirable to briefly describe the composition of the medium used, since the reference is over 50 years old.

A brief recipe was added on lines 310 -315

342, 425 – Please spell out what does IMR stand for (Internet lists 46 possibilities)

(Now lines 309 & 399) We could find no reference to what IMR actually stands for so the name was amended to “enriched seawater medium.”

515-538 – The table 1 looks more like a Figure with an enormous caption, but it is an important contribution and should be in the main text. Suggestion: Transfer the Table in the main part of the contribution and incorporate its “caption” into the main text.

(Now lines 492-504) We agree, unfortunately due to word and figure limitations it must stay in the supplemental materials. It is adequately referenced in the main text though.

537 – Change: Chitons and mollusks. To read: Chiton and other mollusks.

(Now line 503) Much better thanks, fixed.

Respectfully submitted,

Stjepko Golubic

Reviewers' comments:

Reviewer #1 (Remarks to the Author):

My previous comments and concerns have been sufficiently addressed.

Reviewer #2 (Remarks to the Author):

The authors have replied some of my questions in their rebuttal letter but have not made any significant change in the manuscript to address these comments. Therefore, I have quite similar comments as before when reading the revised version. I will try to rephrase as my comments may have not been clear enough in the first review. Of course, these are my comments/questions but as a reader/reviewer I would also expect some related answers showing up in a revised manuscript.

1) Regarding the phrasing, I still think that there is some potential confusion that may rise from the title and several sentences in the manuscript about the idea that solid C may be directly used/transferred to the cells. I am glad that authors do not intend to write that so this should be easy to fix. Here are some sentences:

L17 "suggest fixation of solid carbonate is possible"

L33-34: only C from the dissolved and gaseous pools is a known substrate for autotrophs

L85 "If the sole C source for endolithic biomass had been DIC" and so on...

I understand that the phrasing may become less rigorous at some point in the manuscript and that you may use shortcuts but at least, you should add a sentence from the beginning that makes clear the fact that you are talking about cyanobacteria fixing DIC coming from the cyanobacteria-promoted dissolution of carbonate minerals. The carbon they fix is DIC but a DIC pool that is different from the DIC resulting from CO₂ dissolution. So L85 is misleading. And the phrasing of other sentences is confusing as well.

2) Another problem here is that we have no clue for these experiments what the saturation index (SI) is. This is really important if one wants to understand what is expected thermodynamically. The authors misinterpreted my request. It is not about the local saturation around cells. I understand that this is difficult if not impossible to quantify. I mean the SI of the bulk solutions. This is important to understand the chemical conditions prevailing in the experiments and to predict the global reactivity of the calcite chips. To what extent cyanobacteria catalyze dissolution further is another issue which I understand is beyond the scope of your study. I had additional questions: is the ability to dig is connected to the distance to saturation of the solution? If DIC decreases, the saturation decreases and therefore thermodynamically speaking it may be easier to dissolve carbonates?

3) You answered to reviewer 1 about this question but I feel like the answer is incomplete and related modifications in the manuscript too light: Biomass in endoliths in labelled experiments: is it biomass that was produced during the experiment? Maybe it was produced before the experiment, hence not surprising that you do not see enrichment. You mention that the initial biomass was low. It is not clear by how much it increased. One may think that the infestation depends on the depth. Therefore, when you claim in your response that the probability is very weak, I cannot make the math. How weak is it? You may be completely right but I do not have the numbers to be sure of this. At least, you could mention this assumption explicitly in the manuscript so that we know it is an assumption.

4) Table S1: could you give some kind of mass balance that would support that? How much of the chip needs to be dissolved? Does it correspond to the endolith volume? I do not understand your answer. Since you know the isotopic composition of the mineral and the initial composition of DIC

in the medium, you can calculate a minimal amount of C released from the carbonate. And convert it into a boring volume.

5) How do you explain such a low fractionation (9.58 per mil) in the Isla de Mona samples while one would expect something like 20 per mil?

Reviewer #3 (Remarks to the Author):

This is the second review by this reviewer. The authors have responded to prior critical remarks. The contribution is a major scientific discovery well documented, which has also hitherto unknown and significant relevance to future chemistry and biology of the oceans with special consideration of limestone coasts and coral reefs. It follows logically previous contributions by the same authors using the same model organisms and model environment. It is a prime example of experimental work performed in the field

Reviewer #2 (Remarks to the Author): The authors have replied some of my questions in their rebuttal letter but have not made any significant change in the manuscript to address these comments. Therefore, I have quite similar comments as before when reading the revised version. I will try to rephrase as my comments may have not been clear enough in the first review. Of course, these are my comments/questions but as a reader/reviewer I would also expect some related answers showing up in a revised manuscript.

1) Regarding the phrasing, I still think that there is some potential confusion that may rise from the title and several sentences in the manuscript about the idea that solid C may be directly used/transferred to the cells. I am glad that authors do not intend to write that so this should be easy to fix. Here are some sentences:

L17 “suggest fixation of solid carbonate is possible”

L33-34: only C from the dissolved and gaseous pools is a known substrate for autotrophs

L85 “If the sole C source for endolithic biomass had been DIC” and so on...

I understand that the phrasing may become less rigorous at some point in the manuscript and that you may use shortcuts but at least, you should add a sentence from the beginning that makes clear the fact that you are talking about cyanobacteria fixing DIC coming from the cyanobacteria-promoted dissolution of carbonate minerals. The carbon they fix is DIC but a DIC pool that is different from the DIC resulting from CO₂ dissolution. So L85 is misleading. And the phrasing of other sentences is confusing as well.

The reviewer is correct in that no solid carbonate is being fixed because a dissolved intermediate step is likely to exist in the interstitial space between the solid and the organism. But note that one could argue *ad absurdum* that DIC is never fixed by any organisms either, in that the dissolved bicarbonate/CO₂ molecules always need to go through a non-solvated phase in the interior of RUBISCO enzyme. We are not sure that such distinctions, while formally correct, add much mechanistic value as to the source of the inorganic carbon fixed. In any event, our point all along was that the C that fixed by our organism was never a part of the **bulk seawater DIC** pool that is environmentally a clearly distinct pool from that of mineral carbon. We now revised the manuscript to satisfy the reviewer's suggestion for descriptive rigor, without necessarily deviating the attention of the reader from the novelty of our finding, by agonizing over short-term intermediaries in the process that never become part of the bulk seawater DIC. We revised some of the sentences around the text to make sure rigor was upheld. New sentences are in red type

2) Another problem here is that we have no clue for these experiments what the saturation index (SI) is. This is really important if one wants to understand what is expected thermodynamically. The authors misinterpreted my request. It is not about the local saturation around cells. I understand that this is difficult if not impossible to quantify. I mean the SI of the bulk solutions. This is important to understand the chemical conditions prevailing in the experiments and to predict the global reactivity of the calcite chips. To what extent cyanobacteria catalyze dissolution further is another issue, which I understand is beyond the scope of your study. I had additional questions: is the ability to dig is connected to the distance to saturation of the solution? If DIC decreases, the saturation decreases and therefore thermodynamically speaking it may be easier to dissolve carbonates?

While we still fail to see exactly what the importance of the saturation index of the bulk phase plays in the interpretation of results, particularly when we stated clearly that the medium we use *is* saturated with respect to calcite and we also indicated bulk a lack of significant calcite dissolution by showing very little isotopic exchange between the chip and medium in the closed-vessel non-inoculated incubations. However, we are cognizant that others may somehow find it useful, and so we have now approximated the SI index for media using standard geochemical models, and report it now in the materials and methods (in red type). As mentioned before, SI at the onset and during incubations was above 0. While an increase in biomass in the closed system experiments did lower DIC levels as evidenced by the lack of detectable headspace CO₂, this was accompanied by a significant increase in pH making it uncertain what the bulk SI would have been at the end of the experiments.

Regarding the new questions: yes, presumably and according to our physiological models, as well as thermodynamics, the higher the degree of super-saturation, the harder it would be for organisms to bore. This has however not been addressed experimentally, particularly to see what the limit of this super-saturation would be that cannot be overcome by the light energy captured and used by the microorganisms. But it would certainly be a worthy effort. To the question if a decrease in DIC would also make it easier for calcite to dissolve, this is predictably so, as long as the pH didn't vary. Ultimately, and according to the mechanisms we have presented elsewhere, the controlling factors would be calcium ion and carbonate ion concentrations. To explore or to discuss these questions/ issues is of course, well beyond the scope of this paper; but we note that there exists interesting implications regarding ocean acidification that are currently being pursued by other groups: ocean acidification and a *de facto* decrease in SI, seems to result in increased bio-erosion rates in model natural systems.

3) You answered to reviewer 1 about this question but I feel like the answer is incomplete and related modifications in the manuscript too light: Biomass in endoliths in labeled experiments: is it biomass that was produced during the experiment? Maybe it was produced before the experiment, hence not surprising that you do not see enrichment. You mention that the initial biomass was low. It is not clear by how much it increased. One may think that the infestation depends on the depth. Therefore, when you claim in your response that the probability is very weak, I cannot make the math. How weak is it? You may be completely right but I do not have the numbers to be sure of this. At least, you could mention this assumption explicitly in the manuscript so that we know it is an assumption.

In our response, we did express our high degree of certainty that the biomass that was measured in the NanoSIMS experiments was newly formed. Again here, rigor is on the side of the reviewer. We perceived his problem scenario to be only remotely possible, so that space constraints and a focus on what we perceived as relevant effectively prevented us from giving it more attention.

We see now that nothing short of addressing this with data will satisfy the reviewer. Unfortunately

we did not have image proof of the levels of initial and final infestation in the chips used for our experiments, so in order to satisfy the demand, we had to repeat the initial experimental procedures so as to obtain some, and do some exact calculations

(this took some time to set up, as these organisms are slow growing, and hence the time elapsed in our response).

The chips used to inoculate the experiments (see Figure, A) typically contained colonies occupying some 3.4% of the chip surface, whereas fully colonized chips used for NanoSIMS ranged around 90% area coverage, with much increased density of filaments (some 4-fold) per unit area and with deeper penetration into the chip (Fig 1B). Initially colonized and final non-colonized areas in the figure are demarcated in yellow. While it is principally possible that the data reported in the Nano SIMS experiments stems from cells that were there before the tracer

was added, the probability of this is vanishingly small. The probability of having chosen a surface cell that was preexisting in a fully grown chip equals that of having chosen a portion of the area that was colonized originally ($A = 3.4/90$), and to have chosen one of the preexisting cells within that area ($B = 1/4$), or ($A \times B = 0.0094$). Because we randomly selected 4 filaments in different fields for purposes of replication, the probability of having chosen 4 filaments that were grown before exposure to the tracer becomes 0.0094^4 , or 8×10^{-9} , or close to one in one-hundred million. This greatly exceeds the standard hypothesis acceptance rates in biology of 5 in one hundred. We now present these calculations in the materials and methods to dispel any potential doubts (in red type), and could add the figure as supplementary, if still deemed necessary.

4) Table S1: could you give some kind of mass balance that would support **that** (what is that)? How much of the chip needs to be dissolved? Does it correspond to the endolith volume? I do not understand your answer. Since you know the isotopic composition of the mineral and the initial composition of DIC in the medium, you can calculate a minimal amount of C released from the carbonate. And convert it into a boring volume.

We think we understand now the source of confusion regarding Table S1. Table S1 was there to show that dissolution of calcite occurs in significant amounts only when chips are incubated in medium *with inoculum of cells*. Isotopic dilution as shown suffices to demonstrate directly this point and *no mass balance is required*. The *only* possible source of isotopically light carbon in this experiment is from the chip, and the extreme isotope dilution *only* occurs in the presence of boring biomass. We now provide such desired mass balance data based on the initial DIC content of the vials, the dilution factor in the DIC $\delta^{13}\text{C}$, the original volume of calcite available, and some approximations of volumetric C content in calcite vs. canonical microbial biomass. It is included in the legend to Table S1 (in red type).

5) How do you explain such a low fractionation (9.58 per mil) in the Isla de Mona samples while one would expect something like 20 per mil?

This was not surprising to us, as it is within the range of biomass fractionations seen in complex photosynthetic microbial communities (see for example van der Meer et al., 2003. Fractionations at the 20 per mil or above are really consistently realized only in cultures for a variety of reasons. In nature, sometimes because of “internal biogeochemical recycling of C”, and commonly because of diffusion limitation, the fractionation signals are weaker than maximal potential. This is thus in fact to be expected in natural complex communities such as the ones that were studying here, with many different types of bacterial colonizing the pore spaces opened by the endoliths (see Couradeau et al. 2017). We now added a sentence to make this point in the main text.

Van der Meer, M. T. J., Schouten, S., Sinninghe Damsté, J. S., de Leeuw, J. W., & Ward, D. M. (2003). Compound-Specific Isotopic Fractionation Patterns Suggest Different Carbon Metabolisms among *Chloroflexus*-Like Bacteria in Hot-Spring Microbial Mats. *Applied and Environmental Microbiology*, 69(10), 6000–6006. <http://doi.org/10.1128/AEM.69.10.6000-6006.2003>

Couradeau, E., Roush, D., Guida, B. S., and Garcia-Pichel, F.: Diversity and mineral substrate preference in endolithic microbial communities from marine intertidal outcrops (Isla de Mona, Puerto Rico), *Biogeosciences*, 14, 311-324, doi:10.5194/bg-14-311-2017, 2017.

REVIEWERS' COMMENTS:

Reviewer #2 (Remarks to the Author):

My former concerns have now been sufficiently addressed and I thank the authors for this. I was much more convinced by the latest response and version of the manuscript